# Data-Efficient Bone Segmentation Using Feature Pyramid- Based SegFormer

**DOI:** 10.3390/s25010081

**Published:** 2024-12-26

**Authors:** Naohiro Masuda, Keiko Ono, Daisuke Tawara, Yusuke Matsuura, Kentaro Sakabe

**Affiliations:** 1Master’s Program in Information and Computer Science, Doshisha University, Kyoto 610-0394, Japan; sakabe.kentaro@mikilab.doshisha.ac.jp; 2Department of Intelligent Information Engineering and Sciences, Doshisha University, Kyoto 610-0394, Japan; kono@mail.doshisha.ac.jp; 3Department of Advanced Science and Technology, Ryukoku University, Kyoto 520-2194, Japan; datawara@rins.ryukoku.ac.jp; 4Department of Orthopedic Surgery, Chiba University, Chiba 260-8677, Japan; y-m-1211@khaki.plala.or.jp

**Keywords:** feature pyramid network, Mask2Former, SegFormer, semantic segmentation, transformer block

## Abstract

The semantic segmentation of bone structures demands pixel-level classification accuracy to create reliable bone models for diagnosis. While Convolutional Neural Networks (CNNs) are commonly used for segmentation, they often struggle with complex shapes due to their focus on texture features and limited ability to incorporate positional information. As orthopedic surgery increasingly requires precise automatic diagnosis, we explored SegFormer, an enhanced Vision Transformer model that better handles spatial awareness in segmentation tasks. However, SegFormer’s effectiveness is typically limited by its need for extensive training data, which is particularly challenging in medical imaging, where obtaining labeled ground truths (GTs) is a costly and resource-intensive process. In this paper, we propose two models and their combination to enable accurate feature extraction from smaller datasets by improving SegFormer. Specifically, these include the data-efficient model, which deepens the hierarchical encoder by adding convolution layers to transformer blocks and increases feature map resolution within transformer blocks, and the FPN-based model, which enhances the decoder through a Feature Pyramid Network (FPN) and attention mechanisms. Testing our model on spine images from the Cancer Imaging Archive and our own hand and wrist dataset, ablation studies confirmed that our modifications outperform the original SegFormer, U-Net, and Mask2Former. These enhancements enable better image feature extraction and more precise object contour detection, which is particularly beneficial for medical imaging applications with limited training data.

## 1. Introduction

Semantic segmentation, which associates a category label with each pixel in an image, is an important task in image processing. In recent years, several methods have been developed for semantic segmentation using convolutional neural networks (CNNs), such as Fully Convolutional Networks (FCNs) [1], SegNet [2], and U-Net [3]. Especially in the medical field, the development of technology to automatically extract bones and tumors from medical images such as MRI and CT images automatically using semantic segmentation has attracted much attention [4,5,6,7]. Studies have shown that CNN-based semantic segmentation can extract regions with simple structures, such as the femur and tumor, with a high degree of accuracy [8,9,10,11,12]. However, this approach does not perform well for complex structures because it does not take into account the positional relationships among objects in an image. Thus, we focused on SegFormer [13], which can extract image detail features by segmenting the image while retaining its positional information of the segmented image. SegFormer is a model for segmentation tasks based on the Vision Transformer [14], which is itself a modification of the Transformer [15], which is used in the field of natural language processing for image recognition tasks [16,17,18]. SegFormer uses a hierarchical transformer encoder that improves segmentation accuracy by extracting both fine high-resolution features and rough low-resolution ones. This method has been very successful in natural image segmentation [13,19], but requires a large amount of training data. High performance is not expected when training images are scarce. For tasks in the medical field, it is not easy to collect enough training images. In fact, SegFormer evaluations on several bone datasets have shown low accuracy.

To enhance feature extraction performance, especially for small datasets, we propose a refined SegFormer architecture. Specifically, we improve the resolution of the feature map at each encoder level, increase the depth of the hierarchical encoder, and introduce a Feature Pyramid Network (FPN) based on self-attention in the decoder [20,21,22]. This multi-faceted approach led to improved semantic segmentation accuracy, particularly in boundary detection.

Finally, we evaluated the performance of the proposed model using multiple spine, hand and wrist, and femur segmentation datasets. Our experimental results show that the proposed method achieves an intersection over union (IoU) score on average of 93.1 on the tested spine images, 93.2 on the test hand and wrist image, and 96.3 on the tested femur images. Moreover, our proposed method achieved Dice coefficient (DSC) scores of 96.6 for tested spine images, 96.4 for tested hand and wrist images, and 98.1 for tested femur images. Through cross-validation and statistical testing, we demonstrated that the proposed method achieved statistically significant improvements in accuracy compared to the baseline model, SegFormer. The main contributions of this paper are as follows:We propose a segmentation method that improves the image feature extraction performance of SegFormer to extract more precise image features even on small datasets. Compared to the conventional SegFormer, our method can extract complicated shapes.We increase the feature map’s resolution and convolution layers for each hierarchical encoder to extract higher-quality image features. In addition, the FPN structure with attention effectively uses position information in the image for decoding.We evaluated the performance of the proposed model using about 270 spine, 420 hand and wrist, and 180 femur images. We achieved a high degree of accuracy compared to previous studies.

## 2. Related Work

Many other semantic segmentation methods have been proposed, with CNN-based methods, in particular, being a major method [23,24,25]. In addition, various models have also been proposed that use contextual information for semantic segmentation, including methods that use image features such as ROI [26] to extract complex region boundaries and existing applications [27] to extract areas that are relatively easy to address. These methods are also used for the semantic segmentation of medical images. Despite the numerous efforts mentioned above, CNN-based methods have yet to achieve highly accurate semantic segmentation for bones with complex geometry [12]. Therefore, a method that applies the Transformer, a deep learning model in the natural language field, to the semantic segmentation tasks has been proposed. It shows higher accuracy than CNN-based methods in natural images. In the following sections, we provide details of the SegFormer and Mask2Former architectures.

SegFormer is a deep learning model modified from the Vision Transformer for segmentation tasks. The Vision Transformer is also an improvement of the Transformer, a deep learning model in the field of natural language processing used for image recognition tasks. The SegFormer architecture is illustrated in Figure 1. There are three image processing steps for input images in SegFormer: pre-processing, feature extraction, and classification. During pre-processing, the input image is divided into patch-sized segments, and each segmented image is converted to a one-dimensional vector. Then, a class token is assigned to the one-dimensional data to encompass the image features obtained through the learning process. Feature extraction is performed using the transformer block feature extractor, as shown in Figure 1. In each transformer block, the similarity between the segmented images is calculated as a feature by computing the inner product between the image vectors’ output by pre-processing using the attention mechanism. Furthermore, a Mix Feed Forward Network (Mix-FFN) is introduced in each transformer block, which performs a 3 × 3 convolution to allow the input image size to be changed according to the image size at the time of prediction. Specifically, it reduces the resolution of the input image by 1/2 for each transformer block, thereby extracting features with resolutions of 1/2, 1/4, 1/8, and 1/16 of the original image. The process of outputting features with multiple resolutions dynamically changes the resolution of the input image to the input size of each transformer block, enabling prediction even when the resolution differs between the training and prediction. Then, by connecting four transformer blocks into a hierarchical structure, both high-resolution and low-resolution features can be extracted to complement the image information. For classification, the feature vector of the input image resulting from the feature extraction is input to All-MLP, a decoder consisting only of MLP, to perform image classification and semantic segmentation.

Mask2Former [28] represents a significant advancement in the field of unified segmentation frameworks, building upon the foundations of instance and panoptic segmentation approaches. Its utilization of transformer-based attention mechanisms is at the core of Mask2Former’s architecture. These attention mechanisms compute the relationship between query, key, and value matrices, allowing the model to focus on the most relevant features for the task at hand. The critical innovation introduced by Mask2Former is its masked attention mechanism. This approach adds an attention mask matrix to the standard attention computation, effectively constraining the attention to only the most relevant regions of the input. Mask2Former can achieve greater computational efficiency and improved segmentation accuracy by selectively attending to the most informative areas. In addition to the masked attention module, Mask2Former also incorporates a sophisticated cross-attention mechanism. This module takes the image features and the query embeddings and applies multiple layers of self-attention and cross-attention to refine the segmentation predictions. This iterative process allows the model to effectively integrate visual information with the query-based guidance, leading to more accurate and coherent segmentation results. Mask2Former’s loss function is designed to address various aspects of segmentation quality comprehensively. It combines classification loss, mask loss, and dice loss, each weighted by their respective coefficients. This multi-faceted loss function ensures that the model optimizes both the region delineation and class prediction accuracy, resulting in high-quality segmentation outputs.

## 3. Proposed Method

We propose a novel segmentation method based on SegFormer to extract bones with complex shapes accurately. Our model incorporates three key modifications to the original SegFormer architecture, as illustrated in Figure 2. Firstly, we increase the resolution of the feature maps in the encoder by adding more convolutional layers. This enables the extraction of more detailed image features. Secondly, we integrate a Feature Pyramid Network (FPN) structure into the decoder. This allows for multi-scale feature fusion, preserving detailed information throughout the decoding process. By combining these enhancements, our model effectively captures and segments intricate bone structures, leading to improved segmentation performance.

### 3.1. Data-Efficient Model

The data-efficient model aims to recognize the boundaries of objects in an image more accurately and to improve the quality of feature maps extracted in each transformer block of a hierarchical encoder, even with a small dataset. The data-efficient model replaces the encoder part of SegFormer with a data-efficient encoder, as shown in Figure 2. The architecture of the data-efficient encoder is shown in Figure 3. There are two major improvements to this model, both designed to address the challenge of feature extraction in limited data scenarios.

First, by increasing the number of convolution layers, more localized and hierarchical features can be extracted. This approach is motivated by recent research in computer vision demonstrating that deeper convolutional architectures can capture increasingly abstract and contextually rich representations. The model increases the number of convolutions performed in the Mix-FFN of each transformer block in the hierarchical encoder. Specifically, the number of 3 × 3 convolution layers in each transformer block was increased from one to five, allowing for more precise capture of the bony contours and subtle structural details of the complex. The rationale behind this modification is grounded in the principle of receptive field expansion. As convolutional layers increase in number, the network’s ability to aggregate contextual information improves, enabling a more comprehensive feature representation. This is particularly crucial in small sample datasets, where the model must extract maximally informative features from limited training data.

Second, we propose a novel approach to feature map resolution that addresses the limitations of traditional hierarchical feature extraction. In previous studies, feature maps were typically downsampled to resolutions of 64 × 64 [px], 32 × 32 [px], 16 × 16 [px], and 8 × 8 [px], corresponding to 1/2, 1/4, 1/8, and 1/16 of the input image resolution. Our approach fundamentally reimagines this strategy by introducing higher-resolution feature maps: 256 × 256 [px], 128 × 128 [px], 32 × 32 [px], and 16 × 16 [px], corresponding to resolutions of 1/1, 1/2, 1/4, and 1/8 of the original image. This resolution enhancement is theoretically supported by multi-scale feature learning principles. By maintaining higher-resolution feature maps, we enable the model to preserve more spatial information and fine-grained details throughout the feature extraction process. The preservation of high-resolution feature maps is particularly beneficial in small sample scenarios, where each pixel’s contextual information becomes critical for accurate representation learning. We set the maximum feature map resolution to 256 × 256 [px] to match the original input image resolution, ensuring a comprehensive and detailed feature representation. This approach allows for a more nuanced capture of structural variations, especially in scenarios with limited training data, where each feature becomes increasingly valuable.

### 3.2. FPN-Based Model

The FPN-based model enhances medical image segmentation, particularly for complex structures such as hands and blood vessels, which is crucial to accurately capturing fine-grained anatomical details and broader imaging context. The innovation of this model lies in improving the SegFormer architecture by replacing its decoder with an FPN-based decoder in Figure 3, which integrates a Feature Pyramid Network (FPN) with attention mechanisms. The FPN structure is a pivotal element, facilitating the effective fusion of multi-scale features by combining high-level semantic information with low-level detailed features. Our model enhances SegFormer by integrating an FPN structure into its decoder. This hierarchical architecture facilitates multi-scale feature fusion, combining high-level semantic information with low-level detailed features. This fusion process enables more precise segmentation of complex anatomical structures.

Moreover, a key innovation in our approach is the integration of self-attention mechanisms at each hierarchical level of the decoder. The decoding process follows a hierarchical approach aligned with the FPN structure. Initially, feature maps from the final transformer block undergo 3 × 3 convolution followed by self-attention processing. The resulting features are then upsampled and combined with similarly processed features (through 3 × 3 convolution) from earlier transformer blocks. Each combined feature representation undergoes self-attention processing to capture long-range dependencies. This sequential feature combination and refinement process progresses through the FPN structure, enabling effective multi-scale feature integration. These architectural improvements collectively enhance the model’s ability to handle challenging medical image segmentation tasks, particularly those involving complex anatomical relationships and multi-scale structures.

## 4. Experimental Evaluations and Discussion

We evaluated the proposed model using CT data of the spine and femur provided by the Cancer Imaging Archive [29], an open dataset, and the hand and wrist data we scanned. The proposed method improves two components of the SegFormer: its data-efficient structure and FPN-based architecture. To verify its performance, we used SegFormer as the baseline, ablation methods, U-Net, DeepLabV3+, and the state-of-the-art Mask2Former as a method for comparison. In our ablation methods, we focused on three key factors: for the data-efficient model, we analyzed the impact of varying the number of convolutional layers and the resolution of feature maps within each transformer block; for the FPN-based model, we evaluated the effect of incorporating or omitting the attention mechanism. To demonstrate the statistical significance of model performance across different extraction methods, we conducted K-fold cross-validation and Friedman tests using each dataset. When the Friedman test indicated significant differences, we further performed the Nemenyi test to identify which models exhibited significant performance differences.

### 4.1. Datasets

Examples of spine, hand, wrist, and femur images and ground truth (GT) are shown in Appendix A. The spine dataset used in this experiment ranged from the neck to the tailbone. The femur dataset used in this experiment ranged from the coxa to the knee. The range of the hand and wrist dataset was from the fingertips to the elbow. The GTs for the spine and femur were verified by an experienced person with over 3 years of experience. The hand and wrist GTs were created by orthopedic surgeons to set the ROI. A total of 276 spine images were divided into three sets with a ratio of 8:1:1 for training, testing, and validation. The spine dataset was augmented to three times the number of pictures using Gaussian noise. This brought the number of spine datasets to 828 images. A total of 186 femur images were also applied to the same data augmentation and dividing. In addition, 472 hand and wrist images were used as training images, and 373 images from another patient were used as test images. The hand and wrist dataset was increased by a factor of 4 using CT images of the hand and wrist taken from four different angles, so it contained 1884 images. These images size was 256 × 256 [px].

### 4.2. Experimental Settings

Throughout all our experiments, we maintained consistent conditions. We experimented with PyTorch version 2.5.1 with CUDA 12.1 on Python 3.10.12 and set the batch size to 4, trained for 20 epochs with a learning rate of 0.001, used cross-entropy as the loss function, and employed Adam as the optimizer. The number of classes was two: the site to be predicted and the background. The performance was evaluated in terms of IoU and DSC. The IoU and DSC indicate the percentage of overlap in the images; a higher value means more overlap in the images. Specifically, the maximum value is 1 if the detected and true regions overlap completely, while the minimum value is 0 if there is no overlap at all. IoU is more stringent in evaluating small overlaps. Minor misalignments or slight differences in prediction boundaries are penalized more heavily. DSC is more lenient and tolerant of small discrepancies between predicted and ground truth segmentations. The equations for the IoU and DSC are as follows:(1)$$\begin{array}{c} \mathrm{IoU}=\frac{TP}{TP+FP+FN}. \end{array}$$
(2)$$\begin{array}{c} \mathrm{DSC}=\frac{TP}{TP+\frac{FP+FN}{2}}. \end{array}$$

The implementation details of the base model and the compared methods are as follows. SegFormer employed the proposed hierarchical transformer-based segmentation model, which is grounded in the original architecture by Li et al. [13] and incorporates an efficient transformer encoder. Specifically, the MiT-B3 variant was utilized. U-Net adhered to the standard U-Net architecture introduced by Ronneberger et al. [3], preserving the canonical encoder–decoder structure with skip connections. Mask2Former implemented the Mask2FormerForUniversalSegmentation model. We employed a pre-trained model from Hugging Face and conducted fine-tuning on our dataset by modifying the number of output labels to two while maintaining the architectural integrity of the original model. DeepLabV3+ was instantiated using the torchvision library, specifically leveraging the deeplabv3_resnet50 model with a ResNet-50 backbone.

### 4.3. Statistical Validation

For the spine and femur datasets, we conducted 10-fold cross-validation to ensure robust performance estimation and minimize potential bias from dataset partitioning. The Friedman test was subsequently applied to statistically compare the performance of the proposed method and that of the compared methods under the null hypothesis ($H_{0}$) that there are no significant differences in performance among the methods. A 5% significance level (α=0.05) was used to determine whether to reject $H_{0}$. If the Friedman test indicated significant differences (p<0.05), we further performed a Nemenyi test to identify specific method pairs with statistically significant performance differences.

For the hand and wrist dataset, which utilized distinct patient data for training and testing, we performed 2-fold cross-validation. This approach was chosen to account for the dataset’s unique characteristics and to validate the model’s generalizability across different patient groups. Similarly, the Friedman test was used to assess the statistical significance of performance variations between methods, with the same 5% significance level applied. In cases where significant differences were detected, the Nemenyi test was employed to identify the pairs of methods that exhibited statistically significant performance disparities.

### 4.4. Results with the Spine Dataset

#### 4.4.1. Result of Data-Efficient Model

Table 1 shows the IoU when only the number of convolution layers was increased in the data-efficient model experiment, indicating that increasing or decreasing the number of convolution layers in each transformer block affects the accuracy of spine extraction. In this experiment, the highest accuracy was obtained when the number of convolution layers was two. The IoU was improved when the number of convolution layers was increased to two or more, compared to the SegFormer, which has only one convolution layer, suggesting that increasing the number of convolution layers is effective for feature extraction at the boundary between the spine and non-spine. However, the IoU did not change when the number of convolution layers was increased by three or more. The data-efficient model outputs 256 × 256 [px], 128 × 128 [px], 64 × 64 [px], and 32 × 32 [px] feature maps, but when the original resolution was low and abstract, such as 8 × 8 [px], the image information was greatly degraded. We can assume that the convolution is not expected to be effective. In this study, the number of convolution layers of the data-efficient model for spines was set to two from that point on.

Table 2 shows the IoU when only the resolution of the feature map was increased in the data-efficient model experiment, indicating that increasing or decreasing the resolution of the feature map in each transformed block affects the accuracy of spine extraction. As shown in Table 2, the highest accuracy was achieved when progressively reducing feature maps extracted from each transformer block from 256 × 256 [px]. However, at a resolution of 512 × 512 [px], there were minimal performance differences. This can be attributed to the fact that the input image resolution was 256 × 256 [px], and upscaling to 512 × 512 [px] results in forcible image stretching, causing pixel information distortion and loss. Moreover, 256 × 256 [px] appears to capture sufficient detail for effective segmentation. In this study, the resolution of the feature map of the data-efficient model for spines was set to 256 × 256 [px] from that point on.

The IoU values of the each model for each dataset are shown in Table 3. The DSC values of each model for each dataset are shown in Table 4. Graphs of IoU for all spine test images for each model are shown in Figure 4. The results of the extraction around the neck, waist, and pelvis of the spine by each model are shown in Appendix A. Table 3 shows that, in the experiment with the data-efficient model, the IoU values are higher than those of SegFormer. Therefore, it can be said that the higher resolution of the feature map and increased convolution layers are effective in improving the extraction accuracy. Furthermore, our data-efficient model demonstrated superior performance compared to other methods such as Mask2Former. In fact, from Appendix A, it can be observed that the data-efficient model accurately extracts the spine with fewer omissions compared to Segformer, U-Net, and DeepLabV3+. From Table 4, we observe that the DSC is higher than the IoU, which also suggests that our method accurately extracts the target regions with minimal segmentation defects. The data-efficient model appears capable of extracting detailed image features from a small dataset.

#### 4.4.2. Result of FPN-Based Model

Table 5 illustrates the IoU for FPN-based models, differentiating between models with and without attention mechanisms. The results demonstrate that integrating the FPN structure into the SegFormer decoder effectively improves segmentation accuracy. Furthermore, we found that incorporating attention mechanisms in addition to the FPN structure leads to further performance enhancement. The FPN-based model achieved higher accuracy compared to SegFormer, primarily due to the synergistic effect of three key improvements: the integration of a top-down pathway through the FPN structure enabling effective multi-scale feature fusion, the incorporation of self-attention mechanisms at each decoder level enhancing spatial relationship modeling, and the sequential refinement of features through 3 × 3 convolutions and attention processing, collectively leading to more precise segmentation, particularly in complex anatomical structures such as the spine. From Table 3 and Table 4, we observe that while the FPN-based model shows a smaller improvement margin compared to data-efficient models, it nonetheless demonstrates a significant enhancement in both IoU and DSC scores relative to the compared methods. In fact, from Appendix A, it can be observed that similar to the data-efficient model, the FPN-based model also accurately extracts the spine with fewer omissions compared to Segformer, U-Net, and DeepLabV3+.

#### 4.4.3. Result of Proposed Model

The IoU was improved by about 10.0 for the proposed model experiment, compared to SegFormer, as shown in Table 3. The DSC was improved by about 9.0 for the proposed model experiment, compared to SegFormer, as shown in Table 4. Both IoU and DSC were also improved compared to the data-efficient and FPN-based models. It also showed an improvement in IoU compared to U-Net and Mask2Former. Figure 4 indicates that the IoU was high for all test images. Therefore, adding a convolutional layer and increasing the resolution of the feature map and FPN structure effectively improves extraction accuracy. As the above demonstrates, the proposed model can extract detailed image features from a small number of dataset and can accurately extract the spine, which is a complex shape, by implementing an effective decoder.

#### 4.4.4. Result of Validation and Test

The performance results of the 10-fold cross-validation are summarized in Table 6. From the result table, the standard deviations of the 10-fold cross-validation for each model are sufficiently small, indicating consistent experimental results despite the limited dataset. Furthermore, the Friedman test yielded a test statistic of $\chi^{2}\left(4\right)=40.00$ (*p* < 0.001), demonstrating statistically significant differences in model performance.

Subsequently, to identify specific performance differences between models, we conducted the Nemenyi post-hoc test. The results revealed significant differences between our proposed model and SegFormer (p=0.037). Additionally, statistically significant differences were found when comparing our proposed model with U-Net $\left(p=2.000\times 10^{-4}\right)$ and DeepLabV3+ $\left(p=1.538\times 10^{-7}\right)$. Notably, no significant differences were observed between our proposed model and Mask2Former (p=0.618).

Although no statistically significant differences were observed between the proposed model and Mask2Former, this can be attributed to their inherently high IoU values. The 10-fold cross-validation results demonstrate subtle performance variations between these two models, suggesting their comparable effectiveness in bone segmentation.

### 4.5. Results of the Hand and Wrist Dataset

#### 4.5.1. Result of Data-Efficient Model

As in the spine dataset experiment, Table 1 shows that IoU was improved by increasing the number of convolution layers. In this experiment, the highest extraction accuracy was found when there were three convolution layers; however, as in the spine experiment, we observed no changes in IoU when the convolution layer was increased beyond a certain level. These results suggest that, again, image abstraction due to excessive convolution hinders the extraction of image features. In this study, the number of convolution layers of the data-efficient model for hand and wrist was set to three from that point on.

As shown in Table 2, similar to previous spine results, the variation in feature map resolution across transformation blocks significantly influences hand and wrist extraction accuracy. Our findings reveal that gradually reducing the feature map resolution from 256 × 256 [px] yielded the highest precision. In addition, comparable performance was observed at a resolution of 512 × 512 [px].

The results of the hand and wrist extraction using each model are shown in Appendix A. Graphs of IoU for all test images for each model are shown in Figure 5. The data-efficient model improved IoU by 23.6, compared to SegFormer. From Appendix A, it can be observed that compared to SegFormer, U-Net, DeepLabV3+, and Mask2Former, the model accurately extracted the hand and wrist with fewer omissions. Particularly, the differences in the palm bone structures are notable, and the model is able to detect the gaps between bones. Therefore, it can be said that, as with the spine, a higher-resolution feature map is effective for accurately extracting the bones of the hand and wrist. As in the spine experiment, increasing the number of convolution layers and increasing the resolution of the feature map are also effective approaches to improving the IoU. Even in the hand and wrist dataset, where bone morphology significantly varies across different anatomical regions, the data-efficient model demonstrates the capability to accurately extract bone shape features. Comparing the accuracy of the data-efficient model and the FPN-based model based on the experimental results of the spine and hand and wrist datasets, the data-efficient model showed higher accuracy. Therefore, for complex structured objects in small datasets, it is considered that more detailed feature extraction is required at the encoder level.

#### 4.5.2. Result of FPN-Based Model

From Table 5, we observe that integrating the FPN structure into the SegFormer decoder and incorporating attention mechanisms further improved performance, not only for spine segmentation but also for hand and wrist segmentation. Furthermore, from Table 3 and Table 4, the FPN-based model demonstrates statistically significant improvements in both IoU and DSC scores compared to SegFormer, suggesting that multi-scale feature fusion can be effectively achieved even in scenarios with substantial anatomical shape variations between bone regions. The results also show that the FPN-based model achieves higher accuracy compared to other methods. Although the FPN-based model shows lower IoU and DSC scores compared to the data-efficient model, Appendix A reveals that the FPN-based model more accurately extracts the fingertip region in the output images when compared to the data-efficient model.

#### 4.5.3. Result of Proposed Model

For the proposed model experiment, the IoU improved by 29.3 compared to the SegFormer, by 2.5 compared to the data-efficient model, and by 4.7 compared to the FPN-based model. It also improved by 5.6 compared to Mask2Former. The same precision improvement was observed in the DSC scores. The results of the extraction around the upper arm, palm, and fingers by each model are shown in Appendix A. Comparing the palm and finger images in Appendix A across each model, the proposed model can accurately extract detailed bone-to-bone gaps with no bone loss compared to U-Net and Mask2Former. Figure 5 shows that the IoU is high for all test images. Furthermore, the proposed model is more accurate in extracting the bones than the data-efficient and FPN-based models. The combination of the two models is particularly effective in improving extraction accuracy.

As the above demonstrates, the proposed model effectively extracts the bones of the hand and wrist, which have complex shapes. In addition, the proposed model also guarantees extraction accuracy in terms of versatility since the training data and test data used in this experiment came from different people. In addition, these results suggest that the proposed approach would be effective in practical applications, such as those encountered in clinical settings.

#### 4.5.4. Result of Validation and Test

The performance results of the 2-fold cross-validation are summarized in Table 7. From the result table, the standard deviations of the 2-fold cross-validation for each model are sufficiently small. Furthermore, the Friedman test yielded a test statistic of $\chi^{2}\left(4\right)=8.00$ (*p*
=0.091), demonstrating no statistically significant differences in model performance.

However, these results should be interpreted cautiously due to limited cross-validation. The hand and wrist experiments used entirely different patient data for training and testing, which prevented the use of traditional cross-validation data-splitting methods. Consequently, only two verification iterations were conducted by swapping the training and testing datasets. Notably, even when interchanging the learning and testing datasets, the small standard deviation suggests that segmentation can be effectively performed on one patient’s data using another patient’s training data. Furthermore, the IoU scores demonstrate a significant difference from other methods, indicating the potential effectiveness of the proposed approach in bone segmentation.

### 4.6. Results of the Femur Dataset

#### 4.6.1. Result of Data-Efficient Model

As in the experiment with other datasets, Table 1 shows that IoU was improved by increasing the number of convolution layers. In this experiment, the highest extraction accuracy was found when there were two convolution layers. The accuracy was highest with two convolutional layers for the spine and femur and with three convolutional layers for the hand and wrist. Therefore, it is considered effective to increase the number of convolutional layers as the target shape becomes more complex and to focus more on the object’s contours.

As shown in Table 2, similar to previous results, the highest accuracy was achieved by gradually reducing the feature map resolution from 256 × 256 [px]. Based on the previous results, it is recommended to adjust the input resolution to match the original image resolution.

The femur extraction results using each model are shown in Appendix A. Additionally, the IoU graphs for each model across all test images are presented in Figure 6. The data-efficient model improved IoU and DSC by about 4.0 compared to SegFormer. From Appendix A, it is evident that compared to SegFormer, U-Net, and DeepLabV3+, our model extracts the femur more accurately with fewer missing or excess parts. Therefore, even for simple structures like the femur, increasing the number of convolutional layers or improving feature map resolution can be an effective approach to enhancing IoU.

#### 4.6.2. Result of FPN-Based Model

From Table 5, it was confirmed that integrating the FPN structure into the SegFormer decoder and incorporating attention mechanisms further improved performance, even for simple structures like the femur. From Table 3 and Table 4, the FPN-based model showed improved accuracy in both IoU and DSC compared to SegFormer. Moreover, specifically for the femur, the FPN-based model demonstrated higher precision than the data-efficient model. This is likely because the femur has a consistent shape, which allowed the feature combination through FPN to have a more significant impact.

#### 4.6.3. Result of Proposed Model

In the experiments with the proposed model, the IoU improved by 7.8 compared to SegFormer and by 11.5 compared to U-Net. However, the difference was not significant with other methods. The DSC showed similar results. Nevertheless, since the proposed model has the highest IoU and DSC, the combination of these two models is effective for further improving accuracy.

#### 4.6.4. Result of Validation and Test

The performance results of the 10-fold cross-validation are summarized in Table 8. From the results table, the standard deviations of the 10-fold cross-validation for each model are sufficiently small. Furthermore, the Friedman test yielded a test statistic of $\chi^{2}\left(4\right)=39.28$ (*p*<0.001), demonstrating statistically significant differences in model performance.

Furthermore, to identify specific performance differences between models, we conducted the Nemenyi test. The results revealed significant differences between our proposed model and SegFormer (p<0.001). Furthermore, statistically significant differences were found when comparing our proposed model with U-Net $\left(p=4.681\times 10^{-7}\right)$. In particular, no significant difference was observed between our proposed model with DeepLabV3+ (p=0.055) and Mask2Former (p=0.789).

No statistically significant difference was observed between the proposed model and DeepLabV3+ or Mask2Former, which is believed to be due to their overall high IoU. Additionally, the p-value with DeepLabV3+ was 0.055, which is close to the significance threshold, and the results of the 10-fold cross-validation reveal subtle performance differences between these models, suggesting equivalent effectiveness in bone segmentation.

## 5. Conclusions

The semantic segmentation of complex bones such as the spine, hand, and wrist remains a difficult and time-consuming task, requiring large computational costs and costly GT image datasets often created manually. SegFormer uses a general CNN segmentation model, but it is challenging when extracting complex shapes such as the vertebrae and hands. In this paper, we propose a novel method that improves on SegFormer from two perspectives, outperforming the SegFormer, U-Net, DeepLabV3+, and Mask2Former for spine, hand, wrist, and femur segmentation and extracting accurate bone regions. In small datasets, the proposed model showed superior performance in image segmentation of the spine and hand and wrist by increasing the number of convolution layers in each transformer block of SegFormer, and realized high-resolution feature maps and effective feature decoding by introducing FPN structure in the decoder. The difference in accuracy between the proposed model and SegFormer was shown to be statistically significant. These results indicate that the proposed model demonstrated greater effectiveness compared to traditional methods in recognizing and extracting the contour features of the target from limited data. In addition, the versatility of the proposed model is high because images from different people were used for the training data and the test data in the hand and wrist dataset.

## 6. Future Works

In future research, we plan to use data from multiple directions as input data to extract more detailed object features using a data-efficient model. Specifically, we are currently using axial images but would like to achieve segmentation by adding sagittal plane image features [30] to them. Moreover, when considering implementation in actual clinical settings, we aim to apply unsupervised and semi-supervised learning techniques to reduce the effort required for creating ground truth labels and to enable the continuous addition of patient data.

## Figures and Tables

**Figure 1 sensors-25-00081-f001:**
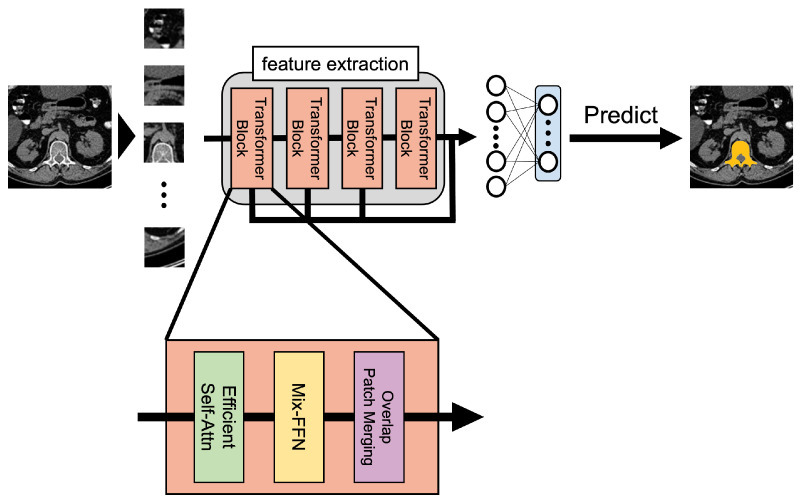
SegFormer architecture.

**Figure 2 sensors-25-00081-f002:**
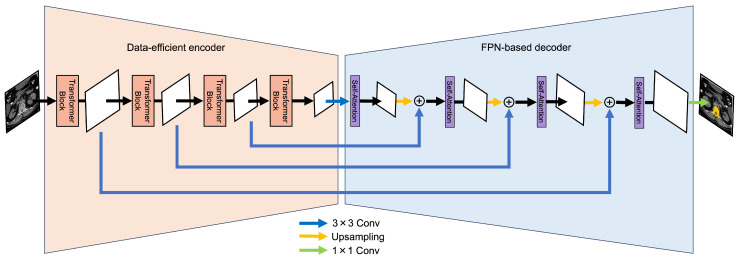
Proposed model architecture.

**Figure 3 sensors-25-00081-f003:**
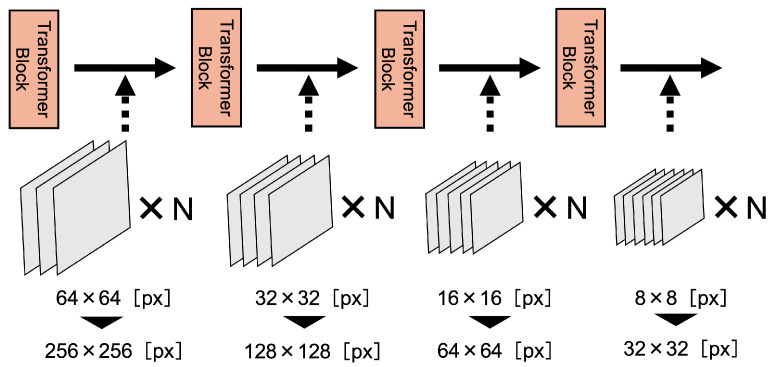
Data-efficient encoder architecture.

**Figure 4 sensors-25-00081-f004:**
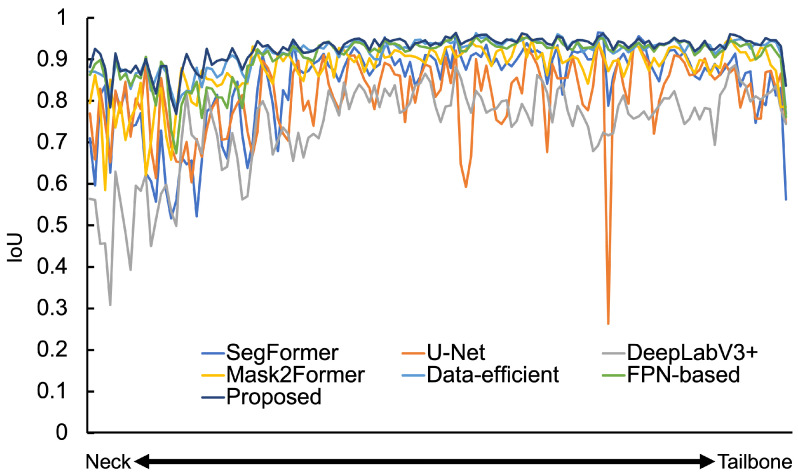
Model-wise IoU for spine images.

**Figure 5 sensors-25-00081-f005:**
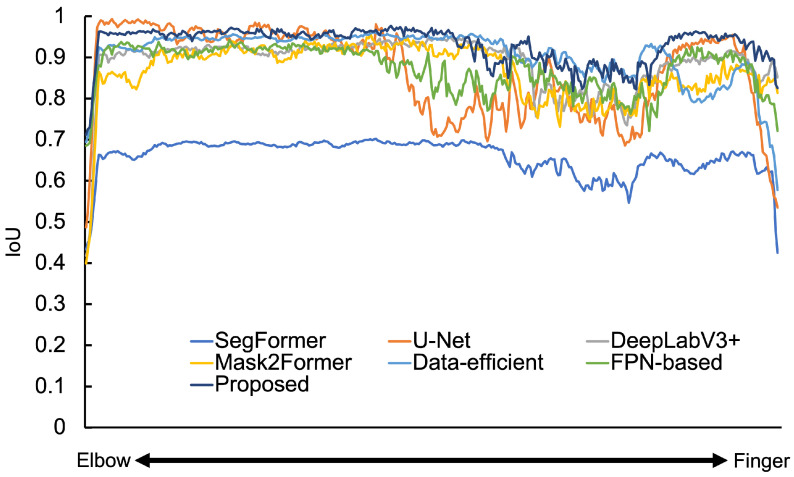
Model-wise IoU for hand and Wrist images.

**Figure 6 sensors-25-00081-f006:**
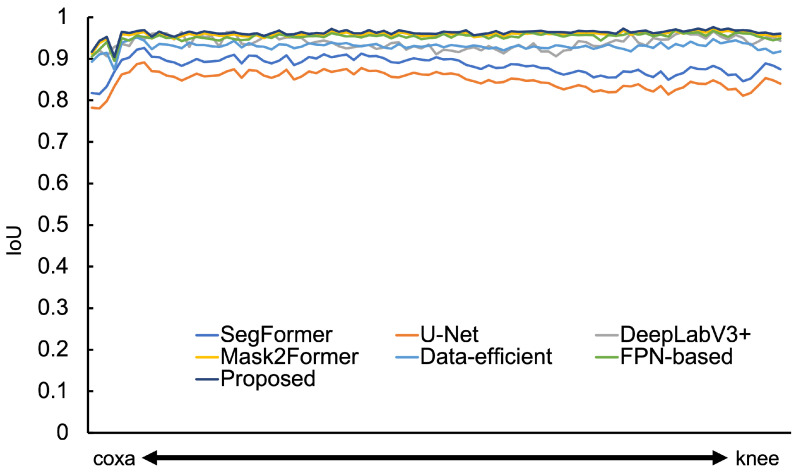
Model-wise IoU for femur images.

**Table 1 sensors-25-00081-t001:** IoU by number of convolution layers.

Number of Convolution Layers	Spine IoU	Hand and Wrist IoU	Femur IoU
1 (SegFormer)	82.7	63.9	88.5
2	**85.9**	68.5	**90.4**
3	85.1	**78.5**	90.1
4	85.2	73.5	89.2
5	84.9	73.2	89.3

Bold values indicate the highest accuracy.

**Table 2 sensors-25-00081-t002:** IoU by resolution of feature map.

Resolution of Feature Map (First Transformer Block)	Spine IoU	Hand and Wrist IoU	Femur IoU
64 × 64 [px] (SegFormer)	82.7	63.9	88.5
128 × 128 [px]	86.2	79.5	90.2
256 × 256 [px]	**89.8**	**84.7**	**91.1**
512 × 512 [px]	89.1	84.1	90.7

Bold values indicate the highest accuracy.

**Table 3 sensors-25-00081-t003:** IoU for each model.

Methods	Spine IoU	Hand and Wrist IoU	Femur IoU
SegFormer	82.7	63.9	88.5
U-Net	76.5	84.3	84.8
DeepLabV3+	74.1	88.9	93.9
Mask2Former	89.1	87.6	95.3
Data-efficient model	91.2	90.7	92.8
FPN-based model	90.7	88.5	95.4
**Proposed Model**	**93.1**	**93.2**	**96.3**

Bold values indicate the highest accuracy.

**Table 4 sensors-25-00081-t004:** DSC for each model.

Methods	Spine DSC	Hand and Wrist DSC	Femur DSC
SegFormer	87.3	73.8	92.1
U-Net	89.4	90.7	89.4
DeepLabV3+	84.5	93.6	96.8
Mask2Former	93.3	92.6	97.6
Data-efficient model	95.1	94.9	96.2
FPN-based model	94.6	92.3	97.6
**Proposed Model**	**96.6**	**96.4**	**98.1**

**Table 5 sensors-25-00081-t005:** IoU by attention mechanism.

IoU by Attention Mechanism	Spine IoU	Hand and Wrist IoU	Femur IoU
SegFormer	82.7	63.9	88.5
FPN-based model without Attention mechanism	88.1	86.2	93.9
**FPN-based model**	**90.7**	**88.5**	**95.4**

**Table 6 sensors-25-00081-t006:** IoU of 10-fold cross-validation for each model.

Methods	Average Spine IoU	Standard Deviation
SegFormer	83.3	0.34
U-Net	77.0	0.46
DeepLabV3+	74.4	0.34
Mask2Former	88.9	0.71
Proposed Model	93.3	0.47

**Table 7 sensors-25-00081-t007:** IoU of 2-fold cross-validation for each model.

Methods	Average Hand and Wrist IoU	Standard Deviation
SegFormer	64.3	0.40
U-Net	83.6	0.70
DeepLabV3+	88.3	0.55
Mask2Former	87.0	0.60
Proposed Model	92.9	0.25

**Table 8 sensors-25-00081-t008:** IoU of 10-fold cross-validation for each model.

Methods	Average Femur IoU	Standard Deviation
SegFormer	88.6	0.40
U-Net	84.9	0.48
DeepLabV3+	93.1	0.55
Mask2Former	95.4	0.37
Proposed Model	96.3	0.33

## Data Availability

The Cancer Imaging Archive dataset can be found at https://www.cancerimagingarchive.net/ (accessed on 12 November 2024).

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
