# Peer review of "Data-Efficient Bone Segmentation Using Feature Pyramid- Based SegFormer"

_sensors, 2024, doi:10.3390/s25010081_

Round 1
Reviewer 1 Report
Comments and Suggestions for Authors
The article titled “Data-efficient Bone Segmentation using Feature Pyramid-based SegFormer” proposed a SegFormer framework for the semantic segmentation in bone images. To improve clarity and relevance, the authors are suggested to make the following improvements.
1. In the section 4 evaluation and discussion section, the authors do not specify the PyTorch, TensorFlow or any framework used for the model training. This might hinder the reproducibility of the results or for understanding technical details. Moreover, they do not explicitly define "GT" when it is first introduced. For the better reproducibility and understanding for the readers, they are requested to add the sufficient details of framework and abbreviations.
2. The authors have mentioned the SegFormer as baseline and compare it with other models like U-Net and Mas2Former, but do not mention the implementation or architecture of these models. This might lead to confusion for practitioners who wants to replicate the work. Therefore, they are requested to provide sufficient details.
3. The discussion section needs improvement as the ablation studies is mentioned but the authors have not provided detailed results or discussion on contribution of modifications to overall performance of model. Therefore, they should add the thorough details of ablation studies to provide a clearer guidance for future research.
4. They could improve their analysis by including a variety of structures in addition to spine, hand, and wrist datasets. This would improve the robustness of bone segmentation tasks and enhance generalizability of the model.
5. They should add the practical aspects of their research findings to close the gap between practical findings and theoretical work making the research more impactful.
6. Though, authors compare their model with U-Net and Mask2Former, a more extensive comparative analysis with previous state-of-the-art methods could provide insights into the strengths and weaknesses of their approach.
7. A separate section of future work is encouraged to include. Moreover, the authors are suggested to further add suggestions for further enhancements or explorations, such as exploring unsupervised learning techniques could provide a roadmap and directions for future research.
Reviewer 2 Report
Comments and Suggestions for Authors
Summary:
The paper proposes several enhancements to SegFormer, a Vision Transformer-based semantic segmentation model, to improve its performance on medical imaging datasets with limited data. The method incorporates a deeper hierarchical encoder with additional convolution layers, higher-resolution feature maps, and a Feature Pyramid Network (FPN) decoder with self-attention mechanisms. The proposed model outperforms baseline methods like SegFormer, U-Net, and Mask2Former in terms of IoU scores on spine, hand, and wrist CT datasets.
Although this paper tackles an interesting problem in image segmentation with limited data, I believe it is not suitable for publication in its current form. Here are some key weaknesses:
Major Issues:
- Clarity:
The paper could benefit from improved clarity, particularly in the "Evaluations and Discussion" section. The experimental setup, results, and analysis are not clearly separated. I recommend presenting the main results first and then creating separate sections for ablation studies, such as those on the number of convolution layers. Additionally, example images of the spine, hand, and wrist could be moved to the appendix to streamline the main text. - Weak Experimental Results:
Since the dataset is very small, the reported results may suffer from high variance and lack reproducibility. Therefore, the reported results is not reliable and convincing. To address this, I suggest using cross-validation for more robust evaluation and potentially employing statistical testing tools to justify the performance gains of the proposed method. - Lack of Ablation Studies:
This paper proposes several modification to the SegFormer. However, only the impact of increasing the number of convolution layers is studied in this paper. I recommend adding ablation studies for other elements as well, as this would help readers better understand the contributions of each component.
Minor Issues:
- Evaluation Metric:
The paper uses IoU as the evaluation metric, but the Dice coefficient is also a widely used metric in image segmentation tasks. I suggest reporting Dice scores alongside IoU or providing some explanation for why Dice is not included. - Lack of Novelty:
I feel the overall novelty of the paper is incremental, as the proposed method mainly involves architectural tuning of SegFormer. - Details on Baseline Method Performance:
This paper lacks details on how baseline performances are obtained. My question is are the hyperparameters for the baselines carefully tuned for these datasets? I recommend adding more details about how baseline models are trained and summarizing hyperparameters in a table in the appendix.
Reviewer 3 Report
Comments and Suggestions for Authors
This paper presents an improved SegFormer architecture for medical image segmentation to extract more accurate and complicated features even on small sample datasets. However, there are some areas in this paper that require the authors to reconsider to revise. The specific opinions are as follows.
1) The improvements made to SegFormer in this paper include adding more convolutional layers and increasing the resolution of the feature maps in the encoder, as well as incorporating a Feature Pyramid Network in the decoder. These improvements are not particularly novel. Please clarify the differences and advantages about the modules used in this paper. Meanwhile, the abstract and the first paragraph of Section 3 mention three improvements, but only two are detailed in the article. It is necessary to clearly list and describe the improvements in the paper to ensure consistency.
2) It provides a detailed description of related work, especially SegFormer and Mask2Former, but the description o f the proposed method is unclear. First, does Figure 2 represent the architecture of the proposed method? The caption seems incorrect. Secondly, the “Data-efficient Model” mentioned in Section 3.1 is not referenced in Figure 2 or earlier in the paper. In Line 139, it is stated that by increasing the resolution of feature maps, it becomes possible to extract fine-gained features from images. This sentence is ambiguous, and the subsequent detailed operation only changes the size of the original feature map. However, the explanation for why these operations lead to better results in small sample datasets is missing, and there is a lack of theoretical support.
3) This paper only uses two datasets for experimental validation, which is not convincing enough. It would be better to write the methods and corresponding metrics after each symbol in the comparison experiments, and also include the ground truth. These experimental results do not provide evidence that better results can be achieved on small sample datasets, as there is no supporting experimental data. Additionally, the ablation experiments are missing.
4) There are some writing errors, such as Line 36, the FPN in Line 44 which is the first mentioned in the paper without its full English name. Additionally, the caption of Figure 5 consists entirely of “(a)”. Please review the entire manuscript carefully.
Round 2
Reviewer 1 Report
Comments and Suggestions for Authors
Most of my comments are addressed. I recommend acceptance of this article in current form.
Reviewer 2 Report
Comments and Suggestions for Authors
I would like to thank the authors for making the revision. Most of my concerns have been addressed.
However, I have one remaining questions: why was 2-fold cross-validation performed on the hand and wrist dataset? A k value of 2 seems too small and may still result in high variance, which could undermine the reliability of the results.
I also believe the writing of the paper should be double-checked before publishing to improve the overall flow and clarity. For example, in line 199, the sentence starts with "in this ablation study", which feels out of place.
Though I still feel the novelty of the work is incremental, the results and analysis appear sound. I would slightly lean toward acceptance if the authors can address the remaining concerns outlined above.
Reviewer 3 Report
Comments and Suggestions for Authors
Thank the authors for addressing my questions and making revisions based on the problems I have pointed out. The additional work enriches the demonstration of the proposed method. However, I noticed that there is only one reference from 2024. I would like to incorporate more recent studies into the discussion to better support the novelty and contributions of the proposed method. Additionally, the reference formatting needs to be revised.
